# Safety, Efficiency, and Mental Workload in Simulated Teledriving of a Vehicle as Functions of Camera Viewpoint

**DOI:** 10.3390/s24186134

**Published:** 2024-09-23

**Authors:** Oren Musicant, Assaf Botzer, Bar Richmond-Hacham

**Affiliations:** Department of Industrial Engineering & Management, Ariel University, Ariel 4076414, Israel; orenm@ariel.ac.il (O.M.); brichacham@gmail.com (B.R.-H.)

**Keywords:** teledriving, camera viewpoint, driving simulator

## Abstract

Teleoperation services are expected to operate on-road and often in urban areas. In current teleoperation applications, teleoperators gain a higher viewpoint of the environment from a camera on the vehicle’s roof. However, it is unclear how this viewpoint compares to a conventional viewpoint in terms of safety, efficiency, and mental workload. In the current study, teleoperators (*n* = 148) performed driving tasks in a simulated urban environment with a conventional viewpoint (i.e., the simulated camera was positioned inside the vehicle at the height of a driver’s eyes) and a higher viewpoint (the simulated camera was positioned on the vehicle roof). The tasks required negotiating road geometry and other road users. At the end of the session, participants completed the NASA-TLX questionnaire. Results showed that participants completed most tasks faster with the higher viewpoint and reported lower frustration and mental demand. The camera position did not affect collision rates nor the probability of hard braking and steering events. We conclude that a viewpoint from the vehicle roof may improve teleoperation efficiency without compromising driving safety, while also lowering the teleoperators’ mental workload.

## 1. Introduction

Teledriving, the remote control of vehicles, is helpful in multiple scenarios, including, for example, delivering a shared vehicle from one person to another or driving electric vehicles to charging stations [1]. In addition, teledriving can also be of great assistance in the interim phase until autonomous vehicles are better able to handle edge-case scenarios. For example, when it is necessary to violate traffic rules (e.g., the road is blocked and one must cross a separation line to go back) or when a police officer is regulating traffic via hand gestures [2].

In view of the multiple scenarios in which teledriving can be of value, different stakeholders are continuously developing and deploying teleoperated vehicles while addressing key design issues [3,4,5]. Many of these issues pertain to the Human Factors aspects of teledriving (see review in [6]), and those (aspects) are often associated with being able to make correct road decisions.

Essentially, similar to in-vehicle drivers, teledrivers must also negotiate the abundance of static and dynamic hazards in the driving environment, like construction zones and other road users [7,8]. However, unlike in-vehicle drivers, teledrivers learn about their road environment from the video images that are streamed to them from the vehicle’s cameras. These cameras can be positioned at different places on the vehicle (e.g., on the vehicle’s roof) instead of where one’s eyes would normally be if inside the vehicle, and thereby teleoperators can gain a different viewpoint on their environment [9,10].

Previous research compared various alternative viewpoints (such as a bird’s eye and a third-person perspective) to a reference viewpoint—typically that of the driver’s perspective—in different scenarios, such as teledriving through obstacle courses or deciding when to overtake another vehicle. These comparisons encompassed different vehicle types, including army vehicles, and assessed indices like distance estimation, accuracy, and collision rates. However, the findings from these studies cannot be easily used to form expectations about safety and efficiency in current teledriving deployment attempts. In such attempts, the alternative to a regular driver viewpoint is from the vehicle’s roof (not from a drone, for example), and the driving environment is most likely urban (and not a terrain or an obstacle course like in certain previous studies).

In the current study, we focused on a use case of teledriving a passenger vehicle in a simulated urban environment. We compared the viewpoint from the vehicle’s roof (hereafter the ‘teledriver viewpont’) to a viewpoint from where a driver would normally be sitting (hereafter the ‘driver viewpoint’) on key measures of task performance and driving safety. In addition, considering that many navigational decisions depend on representations that are relative to one’s own perspective [11], we compared the effect of the two viewpoints on teledriver mental workload.

## 2. Previous Studies on the Effect of Viewpoint on Driving-Related Indices, and Gaps in the Literature

The literature suggests that different viewpoints can affect operators’ performance. Table 1 below presents a summary of studies on the effects of viewpoint on driving-related indices. To keep the focus on the most relevant studies, we included studies that share basic elements with the current study design, such as comparing at least two viewpoints.

An overview of Table 1 reveals that there is scarce research on the effect of camera viewpoints on driving-related indices. A similar observation has been made by Bernhard et al. (2023) [15], and it is unclear why other factors in teledriving, like the time delay, have gained more attention from the research community in this domain (see a review in [6,18]).

Table 1 also reveals that previous studies have used various camera positions, various indices, and otherwise various study designs, and, unsurprisingly, have yielded different findings. For example, Bernhard and Hecht (2021) [14] and Bernhard et al. (2023) [15] had opposing findings on distance estimations if a camera was positioned lower than the driver’s eyes. Both studies, in turn, cannot be compared to van Erp and Padmos’s (2003) [16] findings from a study in which cameras were positioned on the vehicle’s roof (1.7 m or 2.8 m above the ground). Hence, overall, the current literature does not provide repeated findings on the effects of any type of alternative viewpoint (e.g., from a pole on the vehicle’s roof) that could be used as the basis for estimating the effects of other viewpoints.

A mild exception to the lack of repeated findings in the literature might be the two studies in Table 1 [13,17] in which a viewpoint that encompassed both the controlled agent (a simulated vehicle and a robot, respectively) and the environment led to faster completion times (and in Saakes et al. (2013) [17], there were also fewer collisions). These findings correspond with Moniruzzaman et al.’s (2022) [19] explanation that such viewpoints (also termed third-person views) aid obstacle negotiation by keeping both the obstacles and the vehicle in the teleoperators’ visual field. However, in the context of the current study, we note that these findings are still difficult to generalize to other scenarios, like urban driving, because Saakes et al. (2013) [17] conducted a lab study, and in Pazuchanics (2006) [13] the simulated driving was off-road. We also note that, in the context of the current study, teledrivers will not be able to see their entire vehicle in the context of the environment. Rather, the cameras on the roof will only allow them to view the front of the vehicle in the context of the environment (see Figure 1).

Essentially, none of the studies in Table 1 were designed to reflect current teledriving services [20], autonomous vehicle services [21,22,23], or services that would occasionally require teledriver support [24,25]. Many of the studies in Table 1 were designed for other scenarios, and therefore tested viewpoints that are different from the teledriver viewpoint. In addition, safety indices in previous studies were less valid for evaluating driving safety in on-road driving.

Studies in the transportation domain often evaluate driving safety using safety events, such as hard braking and hard steering. Such events are associated with the risk of being involved in road accidents [26,27,28], but, unlike road accidents, they occur more often and, therefore, can be collected in larger pools for statistical comparisons. We believe that these measures should be used for studying the effects of viewpoint on urban teledriving.

Finally, and quite surprisingly, we note that none of the studies in Table 1 have assessed the effect of viewpoint on the teledriver’s mental workload. This is surprising because many navigational decisions depend on representations from one’s own perspective [11], and therefore an alternative viewpoint—such as from the vehicle’s roof—might occasionally require mental transformations into a driver’s viewpoint. Hence, teledriver and driver viewpoints should be compared with respect to mental workload.

## 3. The Current Study

Teledrivers in prospective teledriving services are expected to have a viewpoint from the vehicle’s roof and to operate on-road and often in urban areas. Yet, there is a lack of empirical comparisons between the prospective viewpoint and the reference driver viewpoint in urban driving (real or simulated). Such comparisons are valuable for estimating how operator performance changes across key driving-related indices. In the current study, participants used a teledriver view as well as a driver view in a simulated vehicle in an urban environment. We compared the viewpoints across efficiency indices, including time to complete a task and frequency of navigation errors, safety indices, including the frequency of hard steering, hard braking, and collisions, and the mental workload of the teledriver according to the NASA-TLX.

## 4. Methods

### 4.1. Participants

The study involved 148 students (80 females) that completed two simulated driving sessions utilizing both driver’s and a teledriver’s viewpoints (see Figure 1). The mean age was 25 (SD = 1.8), and the mean driving experience was 6.5 years (SD = 2.4).

The call for volunteers was well received within the student community, enabling us to gather a larger sample than initially planned (for example, the largest sample size reported in Table 1 is 36). The study received approval from the Ariel University Human Ethics Committee (AU-ENG-OM-20210804), and all participants provided written informed consent prior to the experiment.

### 4.2. Design

We investigated dependent variables from three categories:Efficiency: this category included the task completion times of the entire simulated driving route, as well as of each of the challenges along this route; another efficiency index was the probability of making a navigation error. We instructed participants to follow direction signs to navigate their way along the driving route. They could err at three locations (see Table 2 and Figure 2) and any such error had led to a simulation run termination;Safety: this category included collision rates, steering intensity (lateral acceleration m/s^2^), and maximal braking intensity (negative longitudinal acceleration, m/s^2^);Driver state: this category included the drivers’ reported mental workload. We computed the values of all dependent measures as a function of viewpoint (driver vs. teledriver).

### 4.3. Apparatus and Materials

#### 4.3.1. Questionnaires

Participants completed a demographic questionnaire on their age, gender, and driving experience. In addition, at the end of each driving session, participants rated their subjective experience on the six dimensions of the NASA Task Load Index (NASA-TLX) [29]: mental demand, physical demand, temporal demand, performance, effort, and frustration.

#### 4.3.2. Driving Simulator

We utilized the Cognata simulator (Cognata Technologies, Inc., Rehovot, Israel) to conduct simulations of the driving task. The Cognata simulation environment incorporates various elements—such as traffic lights, traffic signs, and other road users—which allow developers of autonomous vehicles to effectively test and train their driving algorithms. In this study, we focused on evaluating the performance of teledrivers; while the ego vehicle (the vehicle being teleoperated by the participant) was not autonomous, the simulator managed other elements, including the behavior of surrounding road users.

To facilitate the operation of the ego vehicle by a human teledriver, we connected acceleration and braking pedals along with a force feedback steering wheel (Logitech G920, San Jose, CA, USA) to the simulator. The simulator setup included a fixed-based configuration with three 27-inch monitors (GIGABYTE 29″X27″, New Taipei City, Taiwan), providing a forward field of view of 135 degrees (see Figure 1). Similar configurations were employed in recent studies [18,30] that also centered on teledriving tasks.

#### 4.3.3. Camera Viewpoints

Each participant completed the task with two camera viewpoints in a counterbalanced order. Figure 1 depicts the position of the virtual cameras for the driver and teledriver viewpoints (left panel), and also shows screen captures of the view from each camera position (right panel). In the teledriver condition, the camera was positioned on the simulated vehicle’s roof (i.e., overhead view), 2.0 m above the road surface, and oriented slightly downward relative to the horizontal plane (as suggested in Prakash et al., 2023 [31]). Note that the driver viewpoint (bottom right panel), 1.4 m above the road surface, showed more of the horizon than the teledriver viewpoint (top right panel), but less of the stretch of road between the vehicle and the curbs on the right and on the left. The teledrivers would naturally like to avoid hitting these curbs; therefore, seeing more of them would aid them in negotiating the road geometry [19]).

#### 4.3.4. Driving Challenges

Table 2 lists the 12 driving challenges (note that #1 in the table is not a challenge but the starting point) in each experimental session. These included turning, overtaking, and pedestrian-crossing. Figure 2 illustrates the driving route with the driving challenges designated by numbers. Route length was approximately 1000 m on a simulated urban road with two lanes, and the participants drove on it in simulated daylight and clear weather conditions.

### 4.4. Procedure

After signing an informed consent form, participants were invited to participate in our driving simulator study. Upon arrival, the participants received detailed information regarding the procedure. They were instructed to drive as they normally would in real road conditions (e.g., obeying traffic rules, following speed limits, and keeping safe headway). All participants completed one training session before the experimental sessions. The training session included roughly five minutes of driving in an urban environment. Our participants were all experienced drivers, and, therefore, well familiar with a driver viewpoint. For this reason, the training session was conducted with the teledriver viewpoint. The order of the driver view and teledriver view conditions was counterbalanced. At the end of each session, participants completed the NASA-TLX questionnaire.

### 4.5. Statistical Analyses

#### 4.5.1. Collision Rates

To study the effect of viewpoint on collision rates, we fitted Cox’s proportional hazards regression. The formal definition of Cox’s proportional hazards regression is given by
(1)$$R_{i}\left(D\right)=R_{Driver\;view}\left(D\right)e^{\beta_{1}*{Teledriver\;view}_{i}+bi}$$
where $R_{i}\left(D\right)$ represents the risk for a collision at driving distance D for participant i; $R_{Driver\;view}\left(D\right)$ represents the risk for a collision at driving distance D in the reference (Driver view) condition. ${Teledriver\;view}_{i}$ is a dummy variable for the viewpoint (Driver view = 0, Teledriver view = 1) and $\beta_{1}$ is its coefficient. The exponent of the coefficient is therefore the risk (often termed hazard) ratio between the teledriver and driver viewpoints. The term bi is a random effect parameter fitted for participant i. We assume bi is normally distributed (bi ∼ N(0, σ)).

#### 4.5.2. Navigational Errors

We used a logistic mixed-effects model for the logit probability of an error (dependent) given the viewpoint (binary variable) as the fixed effect explanatory variable and the participant as a random effect.

#### 4.5.3. Intensity of Braking and Steering Events

The intensity of braking and steering, two key measures of driving safety [26,27,28], is often estimated by their longitudinal or lateral acceleration (in absolute values), respectively. We explored the relationship between viewpoint and the intensity of braking and steering events with two approaches: The first approach was analyzing the probability of a braking or steering event according to a range of intensity thresholds between 2 and 8 m/s^2^, in increments of 0.2 (see Figure 5 in the Section 5). In other words, we estimated the probability for events of intensity greater than 2 m/s^2^ and then for events of intensity greater than 2.2 m/s^2^, etc., with viewpoint (Driver or Teledriver) as the explanatory variable for the probability that braking/steering would be more intense than a threshold intensity (see Botzer et al., 2019 [32], and Hirsh et al., 2023 [33], for additional examples of threshold-dependent analyses). The analysis was based on the following logistic regression model:(2)$$Ln\frac{E\left(Y_{i>T}\right)}{N-E\left(Y_{i>T}\right)}=\beta_{0,T}+\beta_{1,T}*{Teledriver\;view}_{i}+bi$$
where $E\left(Y_{i>T}\right)$ is the expected count of braking (or steering) events that were more intense than T longitudinal (or lateral for steering) acceleration (m/s^2^), out of the total count (denoted by N) of braking (or steering) events. $\beta_{0,T}$ is the intercept, representing the logit proportion of braking (or steering) events with at least T intensity with a driver viewpoint (the reference group). ${Teledriver\;view}_{i}\;$ is a dummy variable with a value of zero for the driver viewpoint and 1 for the teledriver viewpoint, and $\beta_{1,T}$ is its coefficient. Finally, the term bi is a random effect parameter fitted for participant i, where bi is assumed to be normally distributed (bi ∼ N(0, σ)).

The second approach for exploring the relationship between viewpoint and the intensity of braking and steering was looking at the maximal intensity of braking and steering events according to the driving challenge (see Table 2 and Figure 2 for the challenges). We employed a linear mixed-effects model for the maximal intensity ratio (Ln transformed) between the viewpoints (dependent variable) as a function of the driving challenge (fixed effect term) and the participant (random effect term). The purpose of the random effect term was to control for the repeated observations of the same participants over the driving challenges. We note that for each participant we used data entries for 12 challenges when the driving task was completed, and fewer challenges if a collision (or a navigational error) occurred and the simulation run ended. This means that the number of comparisons was slightly different between challenges and varied between 136 and 148 (the number of participants).

#### 4.5.4. Driving Challenges Completion Time

We analyzed the time it took participants to complete the driving challenges. As in the analysis of maximal braking and steering intensity, we again used a linear mixed-effects model. We estimated the natural logarithm of the ratio between the viewpoints on the time to complete driving challenges as a function of the driving challenge (fixed effect) and the participant (random effect). Here also, we focused on challenges that were completed without interference due to a collision or navigational error.

#### 4.5.5. Mental Workload

To compare the influence of camera viewpoint on mental workload, we applied paired *t* tests to the NASA TLX subscales.

## 5. Results

The results are presented in three parts. First, we present the findings on driving efficiency according to task completion times and navigational errors. Second, we present the findings on safety according to collision rates and braking and steering intensity. Last, we present the results on the mental workload scores on the NASA-TLX.

### 5.1. Driving Efficiency

#### 5.1.1. Completion Times

The participants demonstrated a 14% (CI = [10%, 18%]) (the linear mixed model estimates for the effect of teledriver view on Ln (time to complete) was −0.15, s.e. = 0.02, *t* = −6.32, *p* < 0.001) decrease in time to complete the driving route if using the teledriver viewpoint (mean time = 212.9 s, S.D. = 51 s), compared to the driver viewpoint (mean time = 245.9 s, S.D. = 50.7 s). To further investigate the effect of the viewpoint on completion times, we analyzed the completion times separately for the 12 driving challenges. Figure 3 describes the estimated ratio of completion times between the viewpoints. The estimation was based on a linear mixed model to consider the repeated observations for the same participants across the 12 challenges (see Section 4). The reduction in completion times with the teledriver viewpoint was statistically significant in 8 out of the 12 driving challenges in the session. Note that 2 of the challenges which were not completed faster with the teledriver viewpoint involved a crossing pedestrian (bottom two challenges in Figure 3). The drivers needed to wait until the pedestrian had finished crossing the road, and therefore, the potential for the effect of viewpoint on the completion time of these challenges was limited. The other two challenges with non-significant ratios were in left turns in intersections, in which participants had to make sure they could safely enter the intersection.

#### 5.1.2. Navigational Errors

We instructed participants to follow direction signs. In three locations, they could potentially err (see Figure 2 and Table 2). The teledriver viewpoint was associated with ~40% fewer navigation errors than the driver viewpoint (teledriver viewpoint: 6 errors; driver viewpoint: 10 errors). However, the logistic mixed-effects model for the effect of teledriver view showed that it was not statistically significant (the estimated effect for the teledriver viewpoint was −1.44, s.e. = 0.94, z = −1.526, *p* = 0.127).

### 5.2. Driving Safety

#### 5.2.1. Collision Rates

Overall, there was no significant difference in the collision probability between the teledriver (with 20 collisions) and driver (with 19 collisions) viewpoints (mixed effect logistic model estimate for “teledriver” effect on the logit was 0.2, s.e. = 0.62, z = 0.312, *p* = 0.76). A survival analysis (Figure 4, and also see the Section 4) further corroborated the finding from the logistic regression. Figure 4 presents the probability of survival as a function of the camera viewpoint (coded by color) and the distance along the driving route (on the *x*-axis). The two-sided horizontal arrows in the figure designate the driving challenges, with longer arrows for longer road segments. The main decline in survival rates occurred with both camera viewpoints when overtaking a vehicle or overtaking a work zone. Overall, the collision rates were very similar between the two viewpoints across all challenges. Accordingly, the Cox regression model (see Equation (1) in the Section 4) suggested that there was only an insignificant increase of 8% in the risk for collision with the teledriver viewpoint (estimate = 0.07, s.e. = 0.33, OR = 1.08, CI = [0.57, 2.04]).

#### 5.2.2. Probabilities for Braking and Steering Events

Figure 5 below presents the findings from the model in Equation (2) in the Section 4. The figure shows that, as expected, regardless of the camera viewpoint, more intense braking and steering events (left and right panels, respectively) were less frequent than less intense events (the lines in both the left and the right panel are descending). The figure also shows that there were no significant differences in the probability of braking and steering events between the driver and teledriver viewpoints. This is demonstrated by the overlapping of the confidence intervals (vertical gray and black lines). Hence, the proportions of braking and steering events across a wide range of intensities did not change as a function of the camera viewpoint.

#### 5.2.3. Maximal Braking and Steering Intensities by Driving Challenge

Two additional measures for driving safety that we compared between the two viewpoints were the maximal braking and maximal swerving intensity. Figure 6 below presents the ratio of maximal braking intensity (left panel) and maximal steering intensity (right panel) between the driver and teledriver viewpoints during the various driving challenges (*y*-axis). We note that out of all the driving challenges, we were mainly interested in the ratio of maximal braking and steering events on challenges that required responding to hazards. This is because the rationale for using braking and steering events as indices for driving safety is that stronger braking and steering events might point to evasive maneuvers from hazards that were detected late or not estimated correctly [34,35].

Four driving challenges required responding to hazards (see the last four challenges in Figure 6): overtaking a work zone with oncoming traffic, overtaking a vehicle, pedestrian crossing unexpectedly, and pedestrian crossing. Figure 6 shows that, for these challenges, teledrivers did not need to brake or steer significantly harder with either of the viewpoints. Notably, maximal braking or steering was not stronger with the teledriver viewpoint, despite completing the overtaking of the work zone and the overtaking of the vehicle faster with this viewpoint (see Figure 3). Finally, in one of the challenges—the pedestrian crossing challenge—the maximal braking intensity was significantly lower with the teledriving viewpoint (2.9 m/s^2^ (SD = 3) vs. 3.2 (SD = 2.8); the linear mixed model estimate for the Ln intensity ratio between viewpoints was −0.32 s.e. = 0.146, t = −2.195, *p* = 0.03).

Figure 6 also shows that, in the five challenges that did not require responding to hazards, maximal braking or steering was stronger with the teledriver than with the driver viewpoint. For example, maximal steering was significantly stronger with the teledriver viewpoint during two out of the three right turns, and during the left turn at an intersection. In addition, maximal braking was stronger with the teledriver viewpoint during the two left turns at a stop sign at an intersection (seventh and eighth turns in Figure 6, left panel). These stronger maneuvers with the teledriver viewpoint can be attributed to the faster completion of the tasks (see Figure 3), but not to an underestimation or late detection of hazards (because there were not any).

### 5.3. Mental Workload

Figure 7 presents the mean and corresponding confidence intervals per each NASA TLX subscale. Paired t test comparisons show that participants rated their mental demands and frustration lower with the teledriver viewpoint. No significant differences were found in the other subscales of the NASA-TLX.

## 6. Discussion

We compared a teledriver viewpoint from the vehicle’s roof to a driver viewpoint in simulated urban teledriving with respect to indices of driving efficiency, safety, and mental workload.

### 6.1. Driving Efficiency

Participants completed 11 out of the 12 driving challenges faster with the teledriver than with the driver viewpoint, and this difference was statistically significant for 8 of these challenges (see Figure 3). In 2 (out of 4) challenges that were not completed significantly faster with the teledriver viewpoint, completion times were largely independent of the teleoperators’ speed choices. Instead, they mainly depended on how fast pedestrians crossed the road. The other challenges, which were largely completed faster with the driver viewpoint (8 out of 10), involved either turning or overtaking an obstacle (a work zone or another vehicle).

We demonstrated in Figure 1 in the Section 4 (top right panel) that the teledriver viewpoint allowed participants to see more of the surrounding obstacles (e.g., more of the curbs) and, at the same time, part of their own vehicle. Hence, the teledriver viewpoint possibly allowed participants to form better estimations of the vehicle’s position relative to obstacles, turning trajectories, and possible obstacles on the trajectories. Such an effect of a higher viewpoint than from a driver’s seat was conceptually described by van Erp and Padmos (2003) [16] and Moniruzzaman et al. (2022) [19].

Our findings appear to provide a robust case for faster task completion times with a teledriver viewpoint across a myriad of tasks. In this respect, we further note that participants could potentially have made navigational errors at three points along the simulated driving route. Yet, our analysis below Figure 3 in the Section 5 showed that, despite the faster completion of the tasks, navigational errors were fewer (not statistically significant) by approximately 30% with the teledriver viewpoint.

### 6.2. Driving Safety

In our empirical evaluation of driving safety with a teledriver and driver viewpoint, we analyzed collision rates and steering and braking intensities. We found no significant difference in collision rates between the viewpoints (see Figure 4) and no significant difference in the intensity of braking and steering according to intensity thresholds (see Figure 5). Furthermore, a more detailed analysis in which we looked at each driving challenge separately (see Figure 6) showed that maximal braking was not stronger (nor was steering) with the teledriver viewpoint, neither when teleoperators overtook a work zone while facing oncoming traffic nor when they overtook another vehicle, nor when pedestrians crossed the road. Hence, despite completing most of these tasks faster with the teledriver than with the driver viewpoint, driving safety did not appear to be compromised.

We did find that in several of the turning challenges (in five out of eight, see Figure 6), maximal braking or steering was more intense with the teledriver than with the driver viewpoint (although in one challenge, there was an increase in steering intensity but a decrease in braking intensity). However, this is not surprising because almost all turns were completed faster with a teledriver viewpoint (see Figure 3), and completing turns faster means having greater lateral acceleration by definition, which should in turn be moderated by more intense braking [36]. Hence, we do not associate the greater maximal steering and braking in the turning challenges with a lower safety of the teledriver viewpoint. Rather, we associate them with being able to take the turns faster with such a viewpoint because the trajectory was more visible. Further of note is the fact that, in this respect, greater maximal braking or steering with a teledriver viewpoint was only found in turns, and also that we found no difference in the probability of intense braking or steering when the entire (not just a specific challenge) route was considered (Figure 5). Thus, overall, our findings provide no evidence for differences in driving safety between the teledriver and driver viewpoints, whereas almost all tasks were completed faster with the teledriver viewpoint.

### 6.3. Mental Workload

We referred in the Section 1 to previous findings showing that spatial representations from an alternative viewpoint to one’s own position (non-egocentric representations) must be mentally transformed into egocentric representations for making route-following decisions [11]. Considering these possible mental operations when using a teledriver viewpoint (from the vehicle’s roof) we tested if the teledriver and the driver viewpoints differ in mental workload. Our results showed that, with the teledriver viewpoint, teleoperators provided lower ratings on the frustration and mental demand scales of the NASA-TLX (see Figure 7). We note that the gaps in the NASA TLX scores were never beyond 0.5, while the range of the scale is between 1 and 9. These gaps, even if statistically significant, could be considered minor. One might suggest that the putative increase in mental demands from having to transform a teledriver into driver representations was lower than the decrease in mental demands by other mental operations. For example, we already noted that a teledriver view makes it easier to estimate the distance from objects because both the objects and part of the vehicle are in the video frame. Hence, it is possible that the net effects were in favor of lower mental demands with the teledriver viewpoint.

Notwithstanding the high plausibility of this account, we emphasize that our focus in the current study was not on which underlying mechanisms were at play, but rather on collecting empirical evidence on the effects (or lack thereof) of the different viewpoints. The critical take from our analysis is therefore that despite being less conventional, the teledriver viewpoint did not increase the teleoperators’ mental workload.

### 6.4. Limitations and Recommendations for Future Studies

We used a simulator in this study, which is a valuable and commonly used tool for transportation research. However, it is possible that results from simulator studies would differ from results from on-road driving [37] for various reasons, including the extent of driver engagement or the scale of attentional demands [38,39]. Therefore, our findings need further validation in natural driving environments.

Another limitation is that the research population in this study was students rather than trained and experienced teledrivers. The latter population is not readily accessible and could not be recruited within the time frame of the current research project. Studies in different domains have shown that experts employ different task completion strategies than non-experts [40,41], and this might also be the case with teledriving. Hence, future studies should test if the effects of viewpoint that we have found still hold for expert teledrivers.

Other future studies might look into cognitive mechanisms, the exploration of which was beyond the scope of this study. For example, being able to view both the road and part of the vehicle could certainly have been the reason why challenges were completed faster with the teledriver viewpoint. Yet, some might argue that teledrivers drove faster because higher viewpoints lead to higher confidence [14] and to underestimation of speed. The latter, in turn, may result from at least two cognitive mechanisms: first, lower gradient compression ([42], and see accounts in references [14,15]), and, second, reduced visual flow due to less elements being visible in the environment with the camera tilted downward to capture the road from the roof of the vehicle (see on visual flow [16], and compare the views of the environment between the upper and lower right panels in Figure 1).

Future studies might use eye tracking to test if teleoperators fixate on different places depending on the viewpoint (driver or teledriver). In this way, it would be possible to learn if teleoperators indeed utilize the information that is available to them in one viewpoint but not in the other (e.g., do they alternate their fixation between the edge of the vehicle and the curb if they are using the teledriver viewpoint?). Such insights into teledrivers’ scanning patterns might partially address the question as to why they are maneuvering faster with the viewpoint from the roof. In the current study, we mainly focused on the implications of such a viewpoint on safety and efficiency, because it is already in use in teleoperation applications. Therefore, while being able to show that our findings coincide with several acknowledged cognitive mechanisms, we did not attempt to arbitrate between them.

Testing an already existing camera positioning also entailed not attempting to disentangle possible perceptual effects of the height of the camera and its downward angle. Such disentanglement could conceivably be achieved by sophisticated image processing algorithms that would generate images from a higher position as if the camera was not tilted. However, such images would be different from the actual teledriver view in current settings. Note that, in this respect, practical considerations also played a role in Saakes et al. (2013) [17], who tilted a pole on which a camera had been placed, in Bernhard et al. (2023) [15], who chose a relatively small vertical displacement for their cameras according to the feasible displacement in sedans, and in van Erp and Padmos, (2003) [16], who needed to position the higher camera closer to the rear part of the vehicle roof.

Finally, although the indices that we used in our study are considered key for estimates of driver risk by insurance companies [43,44], there are other key measures of driving safety that our study was not designed to collect: for example, time to collision (TTC) and longitudinal and lateral distance from other road users. Future studies with driving scenarios that include merging into traffic and overtaking other vehicles could provide estimates on these indices for the driver and teledriver viewpoints. Hence, while our study provides an important indication for teledriving safety with a camera on the vehicle roof, a fuller picture could only be achieved with a larger battery of driving safety indices.

## 7. Conclusions and Practical Implications

Almost all tasks were completed faster with the teledriver viewpoint (from the vehicle’s roof) while scores on key indices of driving safety were not compromised, and mental workload, if anything, appeared to decrease. These are encouraging findings, considering that future teleoperation services are envisioned to operate often within urban areas and to utilize a viewpoint from the roof of the vehicle. Notwithstanding the encouraging nature of our results, we emphasize the need for future studies with professional teleoperators, with different tasks, and with on-road driving.

## Figures and Tables

**Figure 1 sensors-24-06134-f001:**
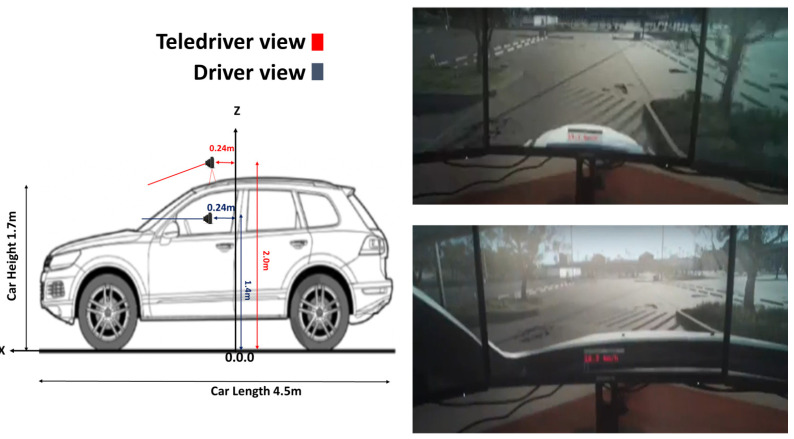
Left panel-position of the simulated cameras. Right panel-Teledriver (top right) and Driver (bottom right) viewpoints on three monitors of 27″ with a forward field of view of 135°.

**Figure 2 sensors-24-06134-f002:**
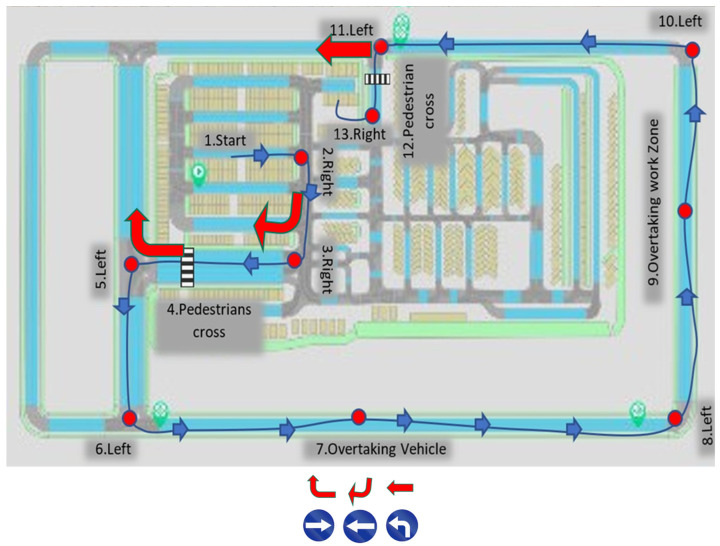
A map of the simulated route and significant points along it. Red arrows indicate three key locations where navigation errors sometimes occurred because participants did not respond correctly to the direction signs (with blue background). These signs are depicted at the bottom of the figure.

**Figure 3 sensors-24-06134-f003:**
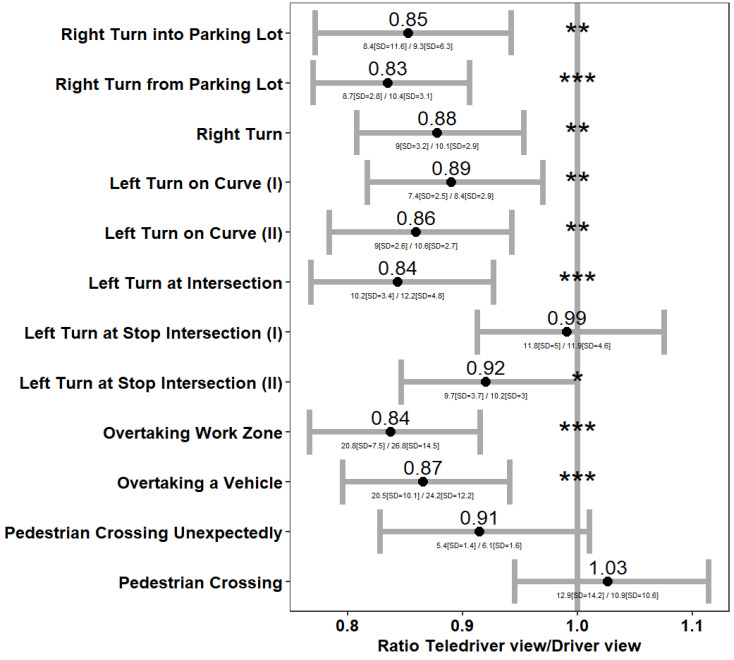
Completion time ratio between the teledriver and driver viewpoints (*x*-axis) by driving challenge (*y*-axis). Notes: (1) The ratio estimates are based on a mixed-effects model to control for repeated observations. (2) Asterisks represent statistical significance: * *p* value < 0.05, ** *p* value < 0.01, *** *p* value < 0.001. (3) Below each confidence interval line, we specify the mean [SD] time (in seconds) to complete the corresponding challenge with the teledriver (in the numerator) and driver (in the denominator) viewpoints. We note that the estimates of the mixed effect model (see note 1) slightly differ from the simple ratio of the means. For example, for pedestrian crossing (last line in Figure 3), the mixed model estimate of 1.03 is different from the ratio of 12.9 s and 10.9 s that we write below it.

**Figure 4 sensors-24-06134-f004:**
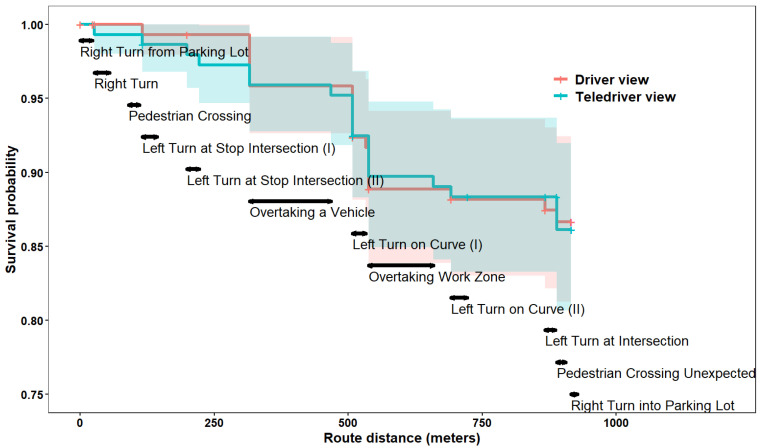
Survival analysis. The survival probability is on the *y*-axis and the route distance is on the *x*-axis. Note: The two-sided horizontal arrows designate the driving challenges, with longer arrows for longer road segments.

**Figure 5 sensors-24-06134-f005:**
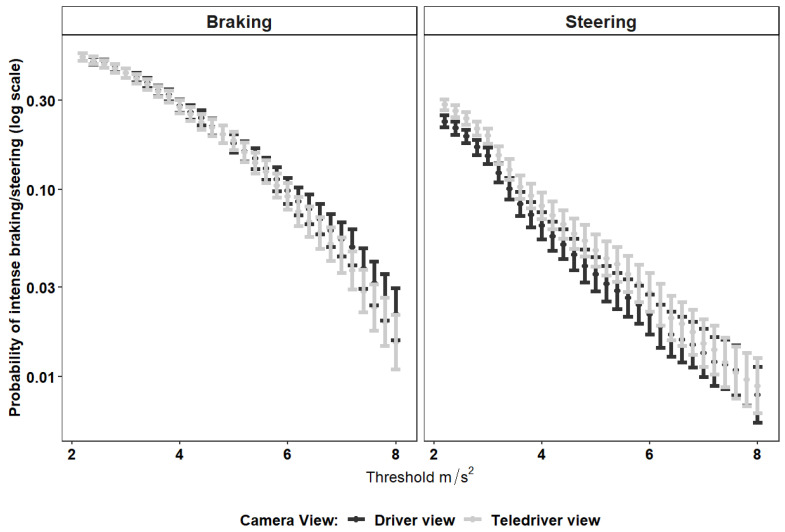
The probability (*y*-axis) of braking events (left panel) and steering events (right panel) as a function of camera viewpoints (separate lines) and acceleration thresholds (ranging from 2 to 8 m/s^2^ on the *x*-axis).

**Figure 6 sensors-24-06134-f006:**
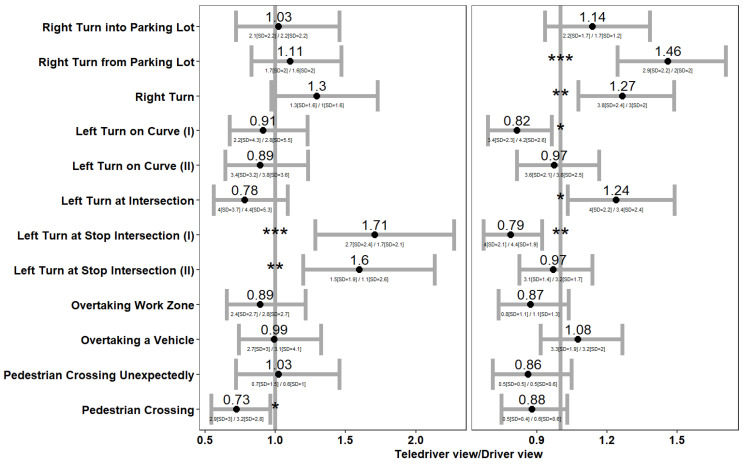
The ratio of maximal braking (left panel) and steering (right panel) intensity between the teledriver and driver viewpoints during the various driving challenges (*y*-axis). Notes: (1) Asterisks represent statistical significance: * *p* value < 0.05, ** *p* value < 0.01, *** *p* value < 0.001. (2) Below each confidence interval line, we specify the mean [SD] of the braking/steering max intensity for the teledriver (in the numerator) and driver (in the denominator) viewpoints. We note that the estimates of the mixed effect model slightly differ from the simple deviation of the means (see a similar note below Figure 3).

**Figure 7 sensors-24-06134-f007:**
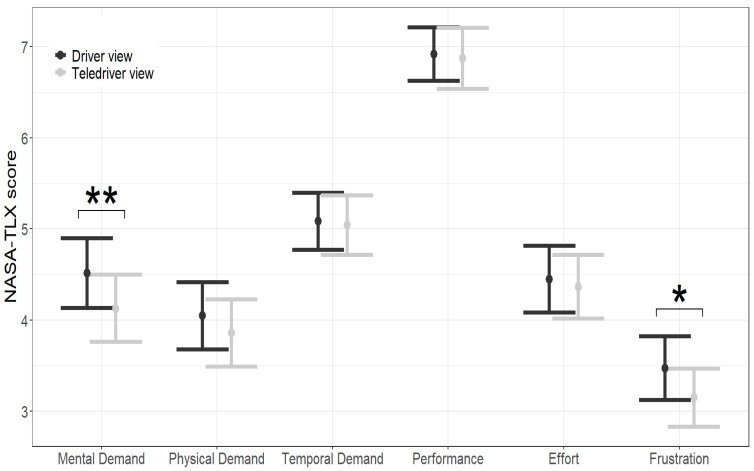
Teledriver and driver viewpoints on the six subscales of the NASA-TLX. Notes: (1) Asterisks represent statistical significance: * *p* value < 0.05, ** *p* value < 0.01.

**Table 1 sensors-24-06134-t001:** Studies on driving-related indices as functions of viewpoint.

Study	Investigated Viewpoint	Vehicle Type	Scenario Characteristics	Indices & Main Findings
Lamb & Hollands (2005) [12]	Egocentric, dynamic tether, rigid tether, three-dimensional exocentric, and two-dimensional exocentric. *n* = 12.	A tank controlled by a joystick.	Off-road navigation through waypoints—Participants had to avoid entering the line of sight of enemy entities.	A. Time to complete the task: faster with the egocentric than with the other displays. B. The amount of time the tank was within sight of any enemy: shorter with the tethered displays than with the other displays and shorter with the exocentric display than with the egocentric display. C. Participant’s recognition of the terrain in which they were driving (Accuracy and time to recognize): Recognition accuracy was best with the exocentric display and better with the tethered displays than with the egocentric display. Recognition time was shorter with the exocentric than with the tethered displays.
Pazuchanics (2006) [13]	First and third-person camera perspective, the latter meaning mounting the camera in a position where it can capture part of the vehicle in the context of its surrounding environment. *n* = 24	Unmanned ground vehicles	Off-road driving through obstacles: A. Tightly zigzagging pathway created with indestructible barriers. B. Pathway created with destructible barriers. C. Field of scattered destructible cars.	A. Time to complete obstacle courses: Faster with a third-person view. B. Number of collisions: No significant effect for camera viewpoint. C. Number of turnarounds (diverging more than 90° from the optimum path for ten seconds.): No significant effect for camera viewpoint. D. Operating comfort: higher with a third-person view.
Bernhard & Hecht (2021) [14]	Exp. 1. Rearview side camera at five positions: conventional and two vertical offsets (low, high) by two horizontal offsets (back, and front). *n* = 20. Exp. 2. Rearview side camera at five positions (very low, low, conventional, high, and very high) with the back of the ego-vehicle either visible or not. *n* = 30.	Passenger Vehicle.	Highway driving scenario.	Accuracy in distance estimation to a following vehicle at different distances from the ego-vehicle: Exp. 1. Low vs. conventional and high camera position resulted in underestimation of distance to a following vehicle. Exp. 2. If the back of the ego-vehicle was invisible: Very low (high) camera resulted in overestimation (underestimation) of distance relative to the conventional camera position. If the back of the ego-vehicle was visible, all the aforementioned effects disappeared.
Bernhard et al. (2023) [15]	Rearview side camera at three vertical positions: Conventional, low (43 cm below conventional) and high (43 cm above conventional). *n* = 36.	Passenger Vehicle.	Highway driving scenario when participants are required to overtake a slow truck. An emerging vehicle visible in the rearview mirror should be considered in an overtaking decision.	A. Safety—participants declared the last safe gap (LSG) for overtaking: LSG was significantly lower for the low camera position as compared to the normal or higher positions. There was no effect for the higher position. B. Self-reported perception of speed, distance, and reward of using the viewpoint: Participants rated the higher camera position as better than the lower position on distance, speed and overall rearview perception. They rated the conventional position as better than the lower position on speed and overall rearview perception.
van Erp & Padmos (2003) [16]	Exp. 1. Two viewpoints on the vehicle roof—1.8 m and 2.8 m above ground. Two FVOs (50° or 100°). *n* = 8. Exp. 2. Two viewpoints (1.8 and 2.8 m) and two FVOs (50° or 100°). *n* = 8.	Passenger Vehicle.	Experiment 1: field test in a close track. Experiment 2: Simulator study.	Exp. 1. A. Lateral control (e.g., course stability) on different routes (e.g., 8 courses, sharp curves, lane change, driving backwards on a curve): There was an interaction between effects of FOV and camera height: In several challenges, with low (High) FOV, performance was better with a low (High) camera viewpoint. B. Longitudinal control (Distance to a stopping line and TTC with respect to a stopping line, difference between actual and target speed): Drivers stopped closer to the line with the higher camera. Exp. 2. A. Lateral control (Sharp curves, lane change): With low (High) FOV, performance was better with a low (High) camera viewpoint (consistent with Exp. 1). B. Longitudinal control (Speed and distance estimation): No significant effects were reported.
Saakes et al. (2013) [17]	Front view from a camera at the rear of the vehicle, bird’s eye view from a camera on a quadcopter, and a downward view from a camera on a pole. *n* = 17.	A robot vehicle.	Search and rescue-navigating through a maze and searching for victims (embodied by toys).	A. Task completion time: faster with a camera on a pole. B. Collision count: was lower with camera on a pole and with camera on a quadcopter. C. Number of victims found: More victims were found with camera on a pole and with camera on a quadcopter.
The current study	Driver vs. teledriver viewpoint from the vehicle’s roof.	Passenger Vehicle.	Simulator study in an urban scenario.	A. Efficiency B. Safety C. Mental Workload

Note: We included field of view (FOV) in the column “Investigated viewpoint” for studies in which FOV interacted with the camera position.

**Table 2 sensors-24-06134-t002:** Description of driving challenges.

The Number on the Map in Figure 2	Action	Road Environment	Challenge Description
1	Start	-	-
2	Turning	Parking lot	Right turn from the parking lot.
3	Turning	Straight road	Right turn. Attention to route directions is required to avoid navigational errors (see red arrow for error).
4	Pedestrian crossing	Crosswalk	Pedestrians crossing
5	Turning	Intersection	Left turn at an intersection with a stop sign. Attention to route directions is required to avoid navigational errors (see red arrow for error).
6	Turning	Intersection	Left turn at an intersection with a stop sign.
7	Overtaking	Oncoming traffic lanes	Overtaking a vehicle with oncoming traffic.
8	Turning	Curved road	Left turn on a curve.
9	Overtaking	Oncoming traffic lanes	Overtaking a work zone with oncoming traffic.
10	Turning	Curved road	Left turn on a curve.
11	Turning	Intersection	Left turn at an intersection. Attention to route directions is required to avoid navigational errors (see red arrow for error).
12	Pedestrian crossing	Crosswalk	Pedestrian cross unexpectedly.
13	Turning	Parking lot	Right turn into the parking lot.

## Data Availability

Data are contained within the article.

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
