# Peer review of "Safety, Efficiency, and Mental Workload in Simulated Teledriving of a Vehicle as Functions of Camera Viewpoint"

_sensors, 2024, doi:10.3390/s24186134_

Round 1
Reviewer 1 Report
Comments and Suggestions for Authors
Review of „Safety, Efficiency, and Mental Workload in Simulated Tele-driving of a Vehicle as Functions of Camera Viewpoint”
The authors have looked at the effect of two camera positions on the efficiency of teledriving. They asked a large number of subjects (N=149) to remotely drive a simulated car in two viewing conditions. Once with a view that a driver would have, were his head bolted to the seat and with eye movements suppressed. And once with a similar view from a camera attached to the roof of the car. Such remote driving or tele-driving is indeed performed in real traffic situations, and an investigation of how to optimize it is a wonderful research topic. The simulated traffic scenarios appear to be sophisticated and state of the art. And the authors have put a commendable amount of effort into their study. The results are clear, the raised camera position is superior to the position at driver’s eye. However, there are a number of serious points that should be fixed before we can recommend publication:
Major points:
The use of the terms of egocentric and allocentric is very misleading. Both perspectives are egocentric. They merely reflect two eye-points at different heights above the ground. An allocentric view would be one where the camera is in a position far away from the would-be driver’s eye, such as a bird’s eye view. In other words, the authors have investigated an effect of camera position rather than an effect of ego- or allocentric vantage point.
The main variable of interest, the visual information best suited for remote driving, is sadly confounded. By changing the location of the virtual camera, three variables have been changed simultaneously:
1 the visibility of an ego reference, i. e. a part of the cockpit or hood of the car
2 the height of the camera
3 the angle of the camera, and thereby the scenery at a distance including the ability to anticipate and the position of the horizon
Thus, it is good to know at the roof camera tilted down is better than the level camera at the driver’s eye-point. But unfortunately, we do not know why it is better. Is it because of the raised position (as Bernhard and Hecht would argue), is it because less distracting car interior detail was available in the raised camera position, or is it because the raised position provided less optical flow.
The latter point is crucial: Subjective speed is always higher for lower vantage points, this could have caused the drivers to drive slower in the case of the lower camera position. This confound can explain most of completion time differences between the two cameras. These confounds need to be discussed. And, importantly, the reader should have a chance to take a closer look by providing descriptive data on driving parameters, such as average speed, for the two camera conditions.
Methods of data analysis:
Descriptive statistics are missing. To eye-ball the results, the reader deserves plain descriptive values (averages, box plots, etc.). Everything is only presented in terms of ratios from the Linear Mixed Model (Time to complete Task, Navigational Errors, Braking/Steering Intensity, NASA-TLX). Especially for the NASA-TLX, it is relevant to see the actual scores on the individual subscales and not just their ratios. Generally, it is unclear why the Linear Mixed Model was used - presumably because of the trips which were terminated due to navigational errors, but this is not addressed in the article.
The number of participants is unusually large. The authors mention that they planned with a smaller number but do not specify what the planned number was. The large number renders small and potentially negligible effects significant, so it is very important to report effect sizes for the statistical tests, or to use tests that supply them, otherwise one would have to be concerned about overinterpretation.
Other points:
The literature review is too bulky just to show that others have looked at navigation-related simulated driving scenarios, and it is too flimsy to be of value to gain good survey information. Either the list should be augmented with the major findings obtained by the various cited studies, maybe by adding a column summarizing these findings. Or the list should be omitted. As is, it is not very helpful. Also note the typo in the Bernhard & Hecht reference.
In the description of the procedure, there is no indication of how long an experimental session approximately lasted. It is also mentioned that the participants had a training session before the experimental sessions, and it would be interesting to know which route/task they trained on, for how long, and whether it was with the low or high camera position.
The authors should elaborate on what they mean by driving safety, which is important since it is the key measure for their comparison. Driving safety here only refers to collisions and hard braking/steering, which may be too narrow. Other, softer factors like adhering to speed limits and lateral/longitudinal distances also affect driving safety over time.
Navigational errors led to termination. It is not mentioned how often this occurred or how the data from these trials were handled. There is frequent mention of statistically (non) significant results, but almost no test statistics are reported.
Author Response
Reviewer_1
The authors have looked at the effect of two camera positions on the efficiency of teledriving. They asked a large number of subjects (N=149) to remotely drive a simulated car in two viewing conditions. Once with a view that a driver would have, were his head bolted to the seat and with eye movements suppressed. And once with a similar view from a camera attached to the roof of the car. Such remote driving or tele-driving is indeed performed in real traffic situations, and an investigation of how to optimize it is a wonderful research topic. The simulated traffic scenarios appear to be sophisticated and state of the art. And the authors have put a commendable amount of effort into their study. The results are clear, the raised camera position is superior to the position at driver’s eye. However, there are a number of serious points that should be fixed before we can recommend publication:
We thank the reviewer for acknowledging the need for our investigation, for acknowledging our efforts in it, and for the positive feedback on our methodology. We also thank the reviewer for all the valuable comments that helped us to improve our manuscript.
Major points:
The use of the terms of egocentric and allocentric is very misleading. Both perspectives are egocentric. They merely reflect two eye-points at different heights above the ground. An allocentric view would be one where the camera is in a position far away from the would-be driver’s eye, such as a bird’s eye view. In other words, the authors have investigated an effect of camera position rather than an effect of ego- or allocentric vantage point.
Response: We agree. Throughout the manuscript, we now use the term “driver view” (instead of “egocentric”) and “teledriver view” (instead of “allocentric”).
The main variable of interest, the visual information best suited for remote driving, is sadly confounded. By changing the location of the virtual camera, three variables have been changed simultaneously:
1 the visibility of an ego reference, i. e. a part of the cockpit or hood of the car
2 the height of the camera
3 the angle of the camera, and thereby the scenery at a distance including the ability to anticipate and the position of the horizon
Thus, it is good to know at the roof camera tilted down is better than the level camera at the driver’s eye-point. But unfortunately, we do not know why it is better. Is it because of the raised position (as Bernhard and Hecht would argue), is it because less distracting car interior detail was available in the raised camera position, or is it because the raised position provided less optical flow.
The latter point is crucial: Subjective speed is always higher for lower vantage points, this could have caused the drivers to drive slower in the case of the lower camera position. This confound can explain most of completion time differences between the two cameras. These confounds need to be discussed. And, importantly, the reader should have a chance to take a closer look by providing descriptive data on driving parameters, such as average speed, for the two camera conditions.
Response: We agree with the reviewer's comment: the differences between the driver’s view and the teledriver’s view involved camera height and angle, and the (non)visibility of the vehicle hood. Accordingly, and following the reviewer suggestion, we now elaborate on this point in the Discussion (3rd and 2nd to last paragraphs of the Discussion):
We explain that in the studies that we have reviewed (detailed in Table 1), changing the camera view often involved at least two factors, such as height and angle because altering just one factor (e.g., height) did not create a feasible alternative view. For instance, placing a camera on a pole in Saakes et al. (2013) required a change in pole (and camera) angle to be effective. Our approach was similar in that we did not attempt to isolate the impact of a single component but to provide a preliminary comparison between the driver’s view and the teledriver’s view on driving efficiency and on indices of driving safety. In addition, we now emphasize to our readers that while the findings that we present on the effect of viewpoint (driver vs teledriver) on driving speed coincide with acknowledged cognitive mechanisms, we do not attempt to arbitrate between these mechanisms. Instead, we show that a teledriver’s view, like that implemented in current teledriving applications, is expected to increase driving efficiency while not compromising performance on several key indices of driving safety.
Regarding the reviewer's suggestion to provide descriptive statistics on driving speed and other driving parameters, the revised version of the manuscript includes the descriptive statistics of our analyses (we address this point in more detail in the response below). Specifically, with respect to average driving speed, we present the route completion times in section 5.1.1, and the route length in section 4.3.4, but we can also report on the average speed if necessary.
Methods of data analysis:
Descriptive statistics are missing. To eye-ball the results, the reader deserves plain descriptive values (averages, box plots, etc.). Everything is only presented in terms of ratios from the Linear Mixed Model (Time to complete Task, Navigational Errors, Braking/Steering Intensity, NASA-TLX). Especially for the NASA-TLX, it is relevant to see the actual scores on the individual subscales and not just their ratios. Generally, it is unclear why the Linear Mixed Model was used - presumably because of the trips which were terminated due to navigational errors, but this is not addressed in the article.
Response: We agree with the reviewer about the missing descriptive statistics:
- Time to complete the task: For the overall driving time throughout the course of our scenario, we have added descriptive statistics (mean and standard deviation) in the text in the first paragraph of section 5.1. In addition, for each of the 12 driving challenges along the route, we have added the descriptive statistics in Figure 3.
- Navigational errors: The count of errors is specified in the last paragraph of section 5.1
- Braking/Steering intensity: For each of the 12 driving challenges along the route, we have added the descriptive statistics for the max braking/steering intensity (m/s^2) in Figure 6.
- NASA-TLX: We accepted the reviewer’s suggestion, and we now present the results from a series of paired t tests. We also replaced the figure (Figure 7) so that it now demonstrates the means (and confidence intervals) of the NASA-TLX scores rather than the gaps between them. The results in terms of significant effects are the same.
The number of participants is unusually large. The authors mention that they planned with a smaller number but do not specify what the planned number was. The large number renders small and potentially negligible effects significant, so it is very important to report effect sizes for the statistical tests, or to use tests that supply them, otherwise one would have to be concerned about overinterpretation.
Response: In the revised Participants section (4.1), in the context of our sample size, we now note in parentheses that the largest sample in Table 1 is 36. We agree with the reviewer that with a larger sample, the meaning of significance might potentially be over interpreted. However, throughout our manuscript, we focus our description of the results and our discussion of them on the mean differences rather than on their statistical significance:
- Time to complete the task (section 5.1.1): Although we do report on statistical significance, we show a ~14% reduction in the time to complete the route, and a reduction of at least 10% for most of the 12 challenges along the driving route.
- Navigational errors (section 5.1.2): This analysis had no risk of over interpretation because the count of errors was not significant even with the large sample size.
- Braking/Steering intensity (sections 5.2.2 and 5.2.3): The gaps in the probability of intense events (section 5.2.2) was largely insignificant (despite the sample size). The analysis of maximal event intensity (section 5.2.3) revealed insignificant results in most of the driving challenges. The few results that were significant showed large effects and we reported changes of over 15% alongside our report on statistical significance.
- NASA-TLX: In two NASA TLX subscales, the gaps were statistically significant but minor. Following the reviewer comment, we now emphasize it in the Discussion in section 6.3 (in the 1st paragraph of this section).
Other points:
The literature review is too bulky just to show that others have looked at navigation-related simulated driving scenarios, and it is too flimsy to be of value to gain good survey information. Either the list should be augmented with the major findings obtained by the various cited studies, maybe by adding a column summarizing these findings. Or the list should be omitted. As is, it is not very helpful. Also note the typo in the Bernhard & Hecht reference.
Response: Following the reviewer’s advice, we augmented Table 1 with the major findings of the studies that we present. We also corrected the typo in Bernhard & Hecht.
In the description of the procedure, there is no indication of how long an experimental session approximately lasted. It is also mentioned that the participants had a training session before the experimental sessions, and it would be interesting to know which route/task they trained on, for how long, and whether it was with the low or high camera position.
Response: We now report on the average completion time of the driving session with each viewpoint (section 5.1.1). In addition, we now describe the training session (duration, environment, viewpoint) in section 4.4. In this context, we also explain that we chose the teledriver view for the training session because our participants were experienced drivers and therefore, were already very familiar with a driver view.
The authors should elaborate on what they mean by driving safety, which is important since it is the key measure for their comparison. Driving safety here only refers to collisions and hard braking/steering, which may be too narrow. Other, softer factors like adhering to speed limits and lateral/longitudinal distances also affect driving safety over time.
Thank you for this comment. In the revised version of the Discussion (last paragraph), we now inform the readers that although the indices that we collected are considered key for estimating driver risk by insurance companies, there are other key indices that our study was not designed to collect. Such indices can be collected in driving scenarios that require merging into traffic and overtaking other vehicles. Therefore, we acknowledge that while our study provides an important indication for teledriving safety with a camera on the vehicle roof, a fuller picture could only be achieved with a larger battery of driving safety indices.
Navigational errors led to termination. It is not mentioned how often this occurred or how the data from these trials were handled. There is frequent mention of statistically (non) significant results, but almost no test statistics are reported.
Response: In the revised version of the manuscript, we report on the number of navigational errors with each viewpoint (section 5.1.2). In addition, we now explain in the last paragraph of section 4.5.3 that participants who were involved in a navigational error or in a crash could not complete all the challenges and that for this reason, some comparisons across driving challenges included 136 data sets instead of 148. And for them, we could use part of the data. We note this matter again in section 4.5.4.
Regarding test statistics, we now include test statistics throughout the revised Results section (for example t or z statistics for mixed effect model estimates). However, we also note that many of our estimates are presented graphically (see Figures 3 and 6) according to the 12 driving challenges. In the revised manuscript, we managed to add means and SDs to these figures and we prefer not to add the test statistics to them as well because we are concerned about reducing their readability. However, if necessary, we can add tables with test statistics (but please consider that this would increase the length of the manuscript).
Additional modifications- We have made an additional modification:
Sample size - The sample size was 148 and not 149. We have found that one of our records was presented twice in the database and removed it. We now report on a sample size of 148 and on test statistics according to this sample. None of our original findings changed following the removal of this single record from our database.

Reviewer 2 Report
Comments and Suggestions for Authors
Dear Editor,
Thank you for inviting me to review the manuscript (ID: sensors-3128223). In the manuscript, the effects of different perspectives on the safety, efficiency, and operator load of remote driving are studied based on simulated driving experiments. Overall, the research results can provide a reference for improving limited autonomous driving. However, there are some defects in the manuscript, which are listed as follows. Therefore, my recommendation is “Minor Revision”.
1. Simulated driving experiments can replace real vehicle driving experiments to some extent. However, the setting of the simulated driving experiment and the details of the subjects are very critical. The authors should add more details of these aspects.
2. The full text is poorly structured. A large number of Class III, Class IV, and non-standard titles are used. The authors need to adjust the structure of the full text and reset the sections to conform to the common norms of academic papers.
3. The size and DPI of all figures are inappropriate.
4. The authors need to combine the Results and Discussion, and comprehensively make a comparative analysis based on the results of this research and the results of previous studies. After that, the authors should give a deterministic research conclusion, preferably in quantitative form.
Sincerely,
Dr. Junyan Han
Comments on the Quality of English LanguageMinor editing of English language required.
Author Response
Reviewer_2
Thank you for inviting me to review the manuscript (ID: sensors-3128223). In the manuscript, the effects of different perspectives on the safety, efficiency, and operator load of remote driving are studied based on simulated driving experiments. Overall, the research results can provide a reference for improving limited autonomous driving. However, there are some defects in the manuscript, which are listed as follows. Therefore, my recommendation is “Minor Revision”.
We thank the reviewer for acknowledging the contribution of our findings to improving limited autonomous driving and for the valuable comments on our manuscript.
- Simulated driving experiments can replace real vehicle driving experiments to some extent. However, the setting of the simulated driving experiment and the details of the subjects are very critical. The authors should add more details of these aspects.
Response: We have added more information about the simulator settings in section 4.3.2. The details about the participants (age, gender, driving experience) are detailed in section 4.1. In addition, we have included more information about the participants' recruitment period (in section 4. 1.). If more information items are required, please list them, and we will be happy to provide these items (if we have them).
- The full text is poorly structured. A large number of Class III, Class IV, and non-standard titles are used. The authors need to adjust the structure of the full text and reset the sections to conform to the common norms of academic papers.
Response: Here are the changes that we applied:
- We have changed the Class III subsections according to Sensors’ word template, and in the revised version of the manuscript, there are no Class IV subsections. We will converge to any additional instructions from sensors’ office if the manuscript is accepted.
- In the Methodology section we moved up the subsection focusing on navigational errors near the section focusing on collisions.
- The size and DPI of all figures are inappropriate.
Response: We have reproduced all figures - we hope that the DPI is correct now. We will converge to any additional instructions from sensors’ office if the manuscript is accepted.
- The authors need to combine the Results and Discussion, and comprehensively make a comparative analysis based on the results of this research and the results of previous studies. After that, the authors should give a deterministic research conclusion, preferably in quantitative form.
We agree with the reviewer approach that a direct comparison with the results of previous studies is often the preferred way for placing findings within context and for honing on the research conclusions. However, it appears that the previous studies to date, were very different from our study in terms of the visual perspectives that they have tested (e.g., camera on a drone; a rearview camera on a Sedan chassis, etc.), and in terms of their dependent variables (LSG; Course instability, etc.). Consequently, we could not offer our readers a direct comparison with the results of the previous studies.
Still, we acknowledge that our readers would like to learn about the findings of the previous studies and therefore, following the reviewer’s comment, we revised Table 1, so it now includes their main findings. In addition, in the revised version of the manuscript, we are doing a better job in communicating to our readers that unlike the current study, the previous studies were not designed to test a camera position that is already in use in practical applications of teledriving (see revised paragraphs above Table 1).
Sincerely,
Dr. Junyan Han
Comments on the Quality of English Language
Minor editing of English language required.
Response: We read the manuscript several times and we believe that we were able to find the sentences that required language editing and to revise them properly. We can edit the manuscript further if the reviewer finds it necessary.
Additional modifications- We have made an additional modification:
Sample size - The sample size was 148 and not 149. We have found that one of our records was presented twice in the database and removed it. We now report on a sample size of 148 and on test statistics according to this sample. None of our original findings changed following the removal of this single record from our database.

Round 2
Reviewer 1 Report
Comments and Suggestions for Authors
Review of revised manuscript "Safety, Efficiency, and Mental Workload in Simulated Teledriving of a Vehicle as Functions of Camera Viewpoint"
We commend the authors on the care with which they modified the manuscript according to our suggestions. We are pleased that they agree with all of our main points. The manuscript has greatly benefited from the changes. We have no further major criticism, however, we have spotted a few minor oversights that should be mended.
The authors state that they “now use the term “driver view” (instead of “egocentric”) and “teledriver view” (instead of “allocentric”). However, the term allocentric is still used in some places, e. g. in the Keywords.
Table 1: the summary of Bernhard et al. 2023 is a bit awkward:
Suggest to change to: “
LSG was significantly lower for the low camera position as compared to the normal or higher positions.
Participants rated the higher camera position as better than the lower position on distance, speed …
Table 1 van Erp & Padmos: likewise awkward, maybe change to:
There was an interaction between effects of FOV and camera height.
Figure 3 seems doubled up – did you forget to eliminate the old figure?
Likewise, Figure 5 is doubled.
Author Response
Comments and Suggestions for Authors
Review of revised manuscript "Safety, Efficiency, and Mental Workload in Simulated Teledriving of a Vehicle as Functions of Camera Viewpoint"
We commend the authors on the care with which they modified the manuscript according to our suggestions. We are pleased that they agree with all of our main points. The manuscript has greatly benefited from the changes. We have no further major criticism, however, we have spotted a few minor oversights that should be mended.
Thank you for acknowledging the care with which we have addressed all the points and suggestions and thank you again for providing us with your points and suggestions that greatly helped us to improve our manuscript. The oversights that the reviewers have spotted in their reading of the revised version were corrected as we detail below. Thank you for your careful reading of our revised manuscript.
The authors state that they “now use the term “driver view” (instead of “egocentric”) and “teledriver view” (instead of “allocentric”). However, the term allocentric is still used in some places, e. g. in the Keywords.
Thank you, we have applied the search function and removed the term allocentric from the manuscript. In the keywords we also removed “egocentric” and substituted “teledriving” for “teleoperation”. In addition, our search revealed a study in Table 1 (Lamb & Hollands, 2005), in which one of the experimental conditions was Egocentric and we referred to it in the summary of the results but not in the description of the experimental settings. This omission had been corrected (see Table 1) and following it, we have read the table and the entire manuscript again to check if there were any additional instances of this (or other) sort. We did not find any additional instances.
Table 1: the summary of Bernhard et al. 2023 is a bit awkward:
Suggest to change to: “
LSG was significantly lower for the low camera position as compared to the normal or higher positions.
Participants rated the higher camera position as better than the lower position on distance, speed …
Thank you, we accepted these suggestions for rephrasing and the summary is now clearer.
Table 1 van Erp & Padmos: likewise awkward, maybe change to:
There was an interaction between effects of FOV and camera height.
Thank you, we accepted this better phrasing.
Figure 3 seems doubled up – did you forget to eliminate the old figure?
Thank you, we removed the unnecessary duplication.
Likewise, Figure 5 is doubled
Thank you, we removed the unnecessary duplication.
